# Pharmacodynamics and Clinical Implications of the Main Bioactive Peptides: A Review

Alessandro Colletti [1,2], Elda Favari [3], Elisa Grandi [1] and Arrigo F. G. Cicero [2,4,*]

1 Department of Science and Drug Technology, University of Turin, 10126 Turin, Italy
2 Italian Nutraceutical Society (SINut), Via Guelfa, 9, 40138 Bologna, Italy
3 Department of Food and Drug, University of Parma, 43124 Parma, Italy
4 I IRCCS S, Orsola-Malpighi University Hospital, 40138 Bologna, Italy
* Correspondence: arrigo.cicero@unibo.it

**Abstract:** Bioactive peptides (BPs) are a heterogeneous class of molecules found in a wide range of plant and animal sources. BPs have a number of different industrial applications including pharmacology (nutraceuticals), food, cosmetology, and pet food. Though BPs were initially used mainly as food additives, today the estimated peptide-based product market is around US $40 billion per year, highlighting consumer demand. The nutraceutical field is one of the most interesting applications for BPs, however there are some limitations to the efficacy of BPs in nutraceutical treatments, including low bioaccessibility and bioavailability. Thus, new extraction and isolation techniques have been developed, using both vegetable and animal sources, to obtain BPs with specific activities and improve the bioactivity and the bioavailability. Randomized clinical trials show a possible relationship between the administration of BPs and the reduction of several cardiovascular risk factors, including hypertension, hypercholesterolemia, hypertriglyceridemia and hyperglycaemia. In addition, BPs exhibit antioxidant, anti-inflammatory, antimicrobial, and anticancer potential, but long-term clinical studies are still needed. The aim of this review is to give a general introduction of BPs, describe their production and application methods, present data regarding bioactivity and bioavailability, and finally highlight the future prospects of this class of molecules in clinical practice.

**Keywords:** bioactive peptides; nutraceuticals; food supplements; functional foods

## 1. Introduction

Bioactive peptides (BPs) are a heterogeneous class of molecules found in a wide range of plant and animal sources [1]. The first bioactive peptide was identified circa 1900 by *Mellander*, who isolated BPs from casein and demonstrated its ability to improve bone calcification in rachitic children [2].

BPs can be defined as peptides between 2 and 20 amino acids able to modulate physiological functions [3]. In general, BP consist of an inactive precursor molecule that becomes active after release of the active site [4] by enzymatic or chemical hydrolysis in the gastrointestinal tract, thus allowing BP to be absorbed through specific peptide transporters [5]. Thus, BP can be classified into exogenous and endogenous molecules, obtained via gastrointestinal digestion or artificially, respectively [6].

Even though BP were initially used mainly as additives for food (in broths, soups, sauces), today the estimated peptide-based product market is around $40 billion per year [7]. BP are used in a number of different fields including pharmacology, food, cosmetology, and pet food. They are typically classified as functional foods, though every country has its own rules [8]. To date, the European regulation provides that there may be an overlap between some molecules that can be commercialized as both drugs and/or food/nutritional supplements. This should also be the case of different BPs. In this regard, an increasingly close dialogue with national and European regulatory authorities is certainly necessary in order to strictly regulate and define the field of BPs and its regulatory area.

One of the most important applications of BP is in the nutraceutical sector. BP regulate many important physiological processes, demonstrating antimicrobial, antioxidant, anti-inflammatory, anticoagulant, anticancer, lipid-lowering, anti-hypertensive, and anti-hyperglycemic effects, most likely through several pathways (depending on their structure and amino acid composition), many of which remain unknown [9]. However, as indicated in Table 1, the level of evidence relating to studies conducted on BPs is not uniform and is highly variable according to the clinical setting.

**Table 1.** Bioactive peptides and clinical applications.

|  | Level of Evidence |
| --- | --- |
| Hypertension | A (Meta-analysis of RCTs) |
| Dyslipidemia | A (Meta-analysis of RCTs) |
| Metabolic Syndrome | C (Single RCT or open label clinical trials) |
| Obesity | C (Single RCT or open label clinical trials) |
| Type 2 diabetes | C (Single RCT or open label clinical trials) |
| Immune dysfunction | D (Preclinical studies, no evidence in humans) |
| Cancer | D (Preclinical studies, no evidence in humans) |
| Pain | D (Preclinical studies, no evidence in humans) |
| Infections | D (Preclinical studies, no evidence in humans) |

RCTs = Randomized clinical trials.

Recently, new extraction and isolation techniques have been developed, using both plant (e.g., bromelain, ficin, papain) and animal (e.g., trypsin, chymotrypsin, pepsin, milk-peptides) matrices, to obtain BP with specific activities to improve bioactivity and bioavailability [10]. The bioactivity and bioavailability of isolated peptides are highly dependent on the degree of hydrolysis during the isolation process. This necessitates both in vitro and in vivo studies of BP applications before their commercialization [11]. Currently, there are more than 1500 BP in the Biopep database, but very few have been studied for clinical applications [12].

The aim of this review is to describe the main animal and plant sources of BP, summarise the production methods, and describe the bioactivity and bioavailability of BP and their limitations in clinical practice. Finally, current and future nutraceutical applications will be discussed.

## 2. Animal Sources

Milk and dairy products are one of the most important sources of BP, which are released during gastrointestinal digestion or food processing, and may be an important in the positive effects of breastfeeding infants in the first months of life [13]. Milk contains proteins with immunomodulatory (such as lactoferrin and immunoglobulins), neurotrophic (opioid peptides derived mostly from the hydrolysis of casein), cytotoxicity, anti-carcinogenic, antibacterial, anti-thrombotic, lipid-lowering, and hypoglycemic activities [14,15]. Both human and bovine colostrum are also a rich source of growth factors and BP, which appear to play a significant role in post-natal development [16]. Endogenous peptides derived from donkey milk (EWFTFLKEAGQGAKDMWR, GQGAKDMWR, REWFTFLK and MPFLKSPIVPF) have been investigated in cardiovascular prevention [17]. The enzymatic proteolysis of whey proteins generates several anti-hypertensive peptides including α-lactorphin (Tyr-Gly-Leu-Phe) and β-lactorphin (Tyr-Leu-Leu-Phe), Tyr-Pro, Lys-Val-Leu-Pro-Val-Pro-Gln, α-lactalbumin and β-lactoglobulin [18]. The use of lactic acid bacteria is a new strategy to obtain BP from the milk fermentation processes. For example, L. *helveticus* LBK-16H-fermented milk contains the BP Val-Pro-Pro and Ile-Pro-Pro, which are well known for their ACE-inhibitory anti-hypertensive molecules [19]. Interestingly, it has been demonstrated that the coupled action of fermentation and in vitro digestion under

gastrointestinal conditions boosts the potential bioactivities of donkey milk, releasing a very rich peptide profile. While assessed in vitro, the bioactivities exhibited by donkey milk fermented makes this product an interesting, potentially functional food [20].

Eggs are another good source of many BP used in medicine and food industry [21]. The peptide Arg-Val-Pro-Ser-Leu from egg white protein exhibits anti-hypertensive activity [22]. More than 60 peptides have been identified in boiled eggs and some of them have demonstrated anti-inflammatory and antioxidant effects in in vitro studies [23].

BP derived from meat products have potential as functional foods and nutraceuticals. Both meat and fish BP exhibit antimicrobial, anti-proliferative and cardiovascular-protective activities in vitro and in vivo, though a limited number of nutraceuticals containing meat-derived BP are commercially available [24]. Arg-Pro-Arg peptide from pork demonstrated the greatest anti-hypertensive activity in vivo when combined with Lys-Ala-Pro-Val-Ala and Pro-Thr-Pro-Val-Pro [25].

## 3. Plant Sources

Plants are also rich in protein and BP. Food processing as well as gastrointestinal hydrolysis of soybean seeds and soy milk generate a variety of BP with established lipid-lowering, anti-hypertensive, anti-cancer, anti-inflammatory, and antimicrobial activity [26]. The fermented soy foods natto and tempeh have been digested with a variety of endoproteases to generate oligopeptides with ACE inhibitory and anti-thrombotic activities [27].

Other rich sources of BP are cereal grains, including wheat, barley, rice, rye, oat, millet, sorghum, and corn. BP from these sources have demonstrated cardiovascular protection (ACE inhibitory peptides, dipeptidyl peptidase inhibitor, peptides with anti-thrombotic, antioxidant and hypotensive activities) [28]. Additionally, cocoa, roasted malt, coffee fermented beer and aged sake contain antioxidant BP; however, clinical studies are still lacking [29].

Vegetable wastes represent an important source of food production by-products and researchers have experimented with new processes for the recovery of BP components from these sources [30]. The process of olive oil extraction generates solid and liquid waste containing proteins and BP hydrolysates (prepared by treatment with different proteases) with numerous biological activities [31]. As another example, an estimated 170,000 tonnes of soybean curd residue (also known as Okara) are produced from 1 million tonnes of dairy-type soymilk (protein content 3.5%) [32]. Okara is 15–30% protein, the low molecular weight fraction (less than 1 kDa) of which consists of digestible bioactive peptides with potent ACE inhibitory and antioxidant activities [33].

## 4. New Sources

There is great interest in identifying new sources of peptides [34]. Nongonierma et al. described the generation of BP from edible insects of the orders Ortophera, Coleoptera, Blattodea, Lepidoptera, Isopera and Hymenoptera. BP derived from these sources demonstrated a large number of biological activities including antimicrobial, antidiabetic, antioxidant, anti-inflammatory, and ACE-inhibitory properties [35]. For example, the peptides KHV, ASL, and GNPWM derived from B. *mori* inhibited angiotensin converting enzyme, reducing both systolic and diastolic blood pressure [36]. BP from crickets (*Gryllodes sigillatus*), mealworms (*Tenebrio molitor*), and locusts (*Schistocerca gregaria*) modulated activity of the enzymes DPP-IV, α-glucosidase, and lipase, but clinical trials have not been conducted [37].

Another source of BP are seafood by-products. BP derived from seafood protein exhibit various bioactivities including antioxidant (e.g., GSGGL, GPGGFI, FIGP peptides from *N. septentrionalis* skin), neuroprotective (e.g., seafood by-product-derived collagen peptides), antidiabetic (e.g., PYSFK, GFGPEL, VGGRP peptides from grass carp skin), ACE inhibitory (e.g., GASSGMPG and LAYA from *G. macrocephalus* skin gelatin via pepsin hydrolysis), DPP-IV inhibitory (e.g., peptides obtained from Atlantic salmon), immunomodulatory (e.g., BP from skipjack tuna bones), antibacterial (e.g., BP from crustaceans), antiproliferative, and anticancer (e.g., hydrolysates prepared from rainbow trout) activities [9].

Finally, different plant seeds are cheap sources of antioxidant (e.g., wild hazelnut peptides), antibacterial (e.g., SMRKPPG identified from peony seed), anticancer (e.g., the protein hydrolysate purified from amaranth seeds) antidiabetic (e.g., BP from wild hazelnut), and anti-hypertensive (e.g., flaxseed-derived peptides) peptides [9].

### 4.1. From Production to Commercialisation

BP can be obtained using several methods, which generally include the extraction and the isolation of proteins from a food source, followed by the isolation of a specific protein isolate that undergoes in vitro or in vivo enzymatic hydrolysis. Different enzymes can be used to generate short-chain peptides. Once the protein hydrolysate is obtained, peptides can be separated and purified, and their bioactivities assayed. Subsequently, the peptide fraction of interest is analyzed to determine the amino acid sequence. Once the peptide sequence is known, BP can be synthetized and studied to determine bioavailability and bioactivity before proceeding with clinical trials (Figure 1) [38]. As a recent application, that shows how the in silico methodologies coupled to in vitro tests may be useful for the identification of potentially bioactive peptides, this study methods has been applied to the evaluation of the overall nutritional value of the Parma ham [39].

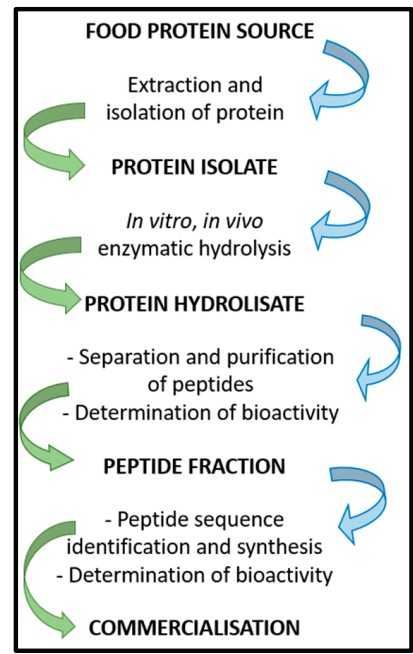

**Figure 1.** Scheme of BP isolation, preparation, and commercialization.

Enzymatic hydrolysis, microbial fermentation, and chemical hydrolysis are the three main methods of BP production. Chemical hydrolysis consists of the formation of peptide hydrolysates using an acid or base at high temperatures resulting in the peptide cleavage. However, though this technique yields robust hydrolysis, it is nonspecific, poorly reproducible, and leads to amino acid denaturation [40]. Enzymatic digestion produces peptides from soy, corn, potato, peanut, milk, whey, egg, and meat proteins with target functionalities including antioxidant, anti-inflammatory, anti-hypertensive, anti-diabetic, anti-microbial, and anti-cancer activities. The efficacy of protein hydrolysates depends on several factors such as the protein substrate pre-treatment, type of proteases used, and the hydrolysis conditions applied [41]. Microbial fermentation produces BP from a variety of sources with a variety of bioactivities. BP produced by microbial fermentation can differ in type, amount and activity depending on the cultures used [42]. Recently, the fermentation of milk proteins with specific microbial strains (e.g., *Lactobacillus bulgaricus*, *Lactobacillus helveticus* MB2-1, *Lactobacillus plantarum* B1-6 and 70810) has produced anti-hypertensive and lipid-lowering peptides [43].

BP with antihypertensive, antioxidant, and antidiabetic activities account for the majority of peptides in the Biopep database, but antimicrobial, antiproliferative, and lipid-lowering activities have also been described [44]. The main limiting factor in the generation of BP is not at the level of raw materials, but the technological processes and the difficulty in efficiently generating a specific BP sequence without altering its functionalities or bioaccessibility [44]. New separation and purification techniques (e.g., ultrafiltration and/or advanced chromatography techniques) have been developed to overcome these problems [45].

### 4.2. Stability and Bioavailability

Once a peptide is ingested, it faces physicochemical environments that can negatively influence its stability and bioactivity. The bioactivities of BP are highly variable and depend on the degree of hydrolysis during the isolation processes, the gastrointestinal environment, BP size, and peptide hydrophobicity [46]. In addition, the food matrix could interact negatively or positively with the chemical structure of BP, modifying both the stability and bioavailability [40]. BP can undergo chemical hydrolysis in the stomach where either acidic conditions or gastric enzymes interact and hydrolyze proteins and peptides. Additionally, pancreatic enzymes, proteases from the microbiota, and the drastic change in pH (from ~2 of the stomach to 7 of the large intestine) can influence the hydrolysis of BP in the gastrointestinal (GI) tract [47]. BP can also be degraded at the brush border, in the cytosol of enterocytes, or even within lysosomes and other cellular organelles [48]. Thus, in vitro studies using simulated gastrointestinal digestion systems are needed to investigate the stability and bioaccessibility of many peptides from food proteins [49]. However, observations from in vitro studies might not be reflective of in vivo conditions, especially for BP with low bioavailability, which is strongly influenced by the absorption and susceptibility to physiological enzymes that breakdown the peptides into inactive fragments [50]. For example, the peptides MAP1 and MAP2 derived from milk proteins demonstrated in vitro ACE inhibitory activity, but only MAP1 has an acceptable stability and can reach the desired cellular sites of action and act as an anti-hypertensive peptide in vivo [51]. Moreover, it has been suggested that it can be useful to develop a delivery coating strategy for the harmless control of ACE inhibitory peptides as a new approach for improving their activity and stability [52].

Bioactive peptides are transported into the blood through several mechanisms. Intestinal peptide transporters can carry smaller peptides across the intestinal epithelium. Paracellular transport (passive transport) is used to move hydrophobic oligopeptides into the bloodstream [53]. Transcytosis is an intestinal transport system by which particles are taken up by cells and subsequently released into the bloodstream [54].

Hayakawa et al. demonstrated that in some areas of the jejunum and ileum the aminopeptidase activity is low compared with other parts of the gastrointestinal tract, constituting a potential targeting sites for BP delivery [55]. Orally administered BPs can encounter over 40 different types of proteases during their passage to the small intestine and over 60 lysosomal peptidases, increasing the risk of degradation and reduction of their bioactivity [56]. Cysteine, basic amino acids, and aromatic amino acids are particularly sensitive to endogenous oxidation and nitrosation processes [57]. In this context, novel strategies (Figure 2) have been developed to improve oral bioavailability of BP, using enzyme inhibitors and/or absorption enhancers (which protect BP from enzymatic degradation and improve their hydrophilicity or surface charge) as well as site-specific release forms to avoid or limit the intestinal enzymatic hydrolysis [58].

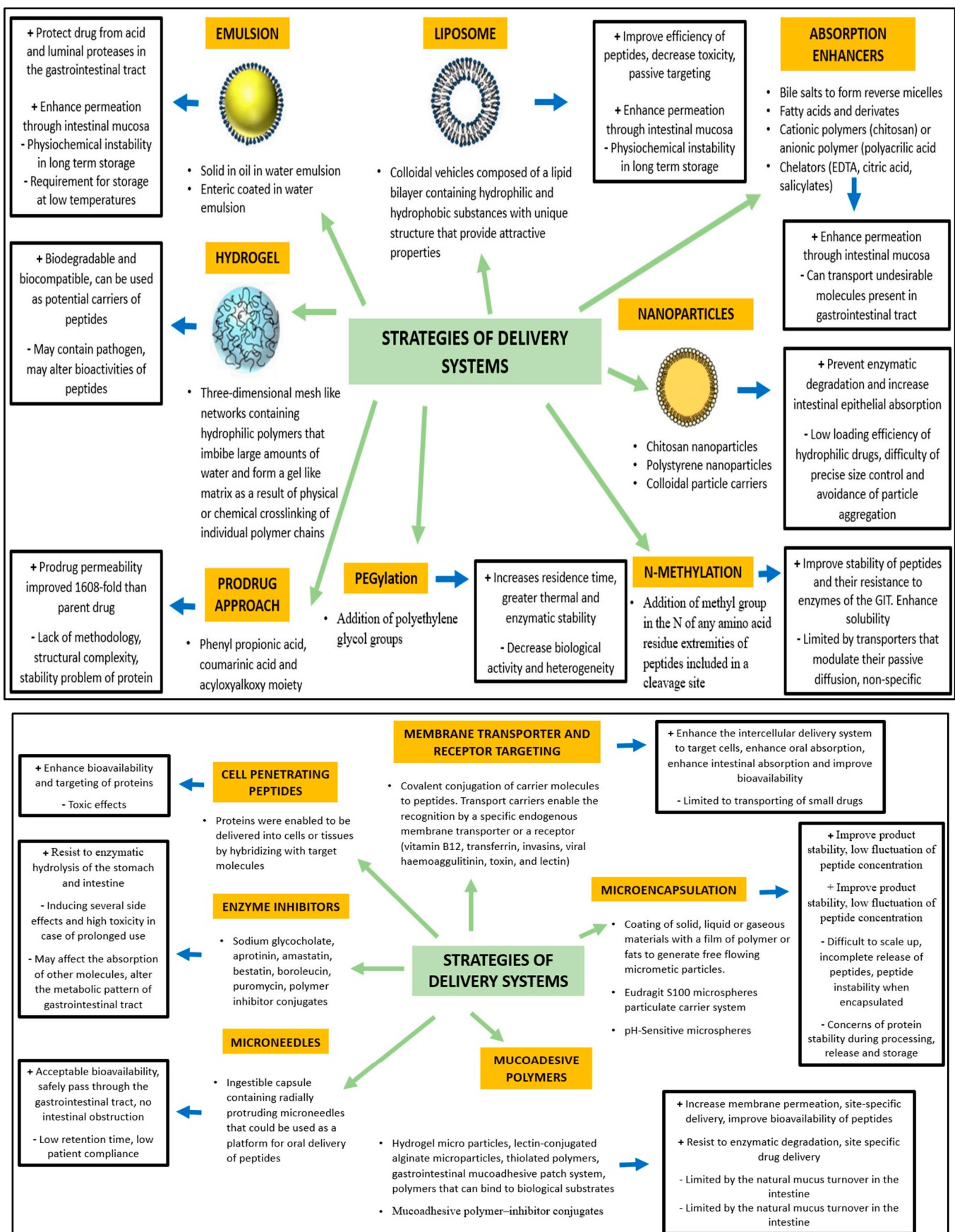

**Figure 2.** Strategies of delivery systems for bioactive peptides (Modified from Bechaux et al. 2019 [44]).

## 5. Applications

The most important application for BP is currently in nutraceuticals [59]. Nutraceuticals are molecules of either animal or plant origin that exert a positive impact on human health when incorporated into food or pharmaceutical formulations (sachets, capsules, tablets, etc.) [24]. Food-based BP are classified into different categories including traditional, or functional, foods (e.g., yogurt, milk, soy, or cheese naturally rich in "substances" such as BP that can be useful for maintaining health), novel foods (foods containing BP that do not have a significant history of consumption and still require safety studies), and fortified foods (foods with added substances such as BP to enhance their nutritional value or physiological effects) [60]. Recently, nutraceuticals and functional foods containing BP have attracted attention for their capacity to improve health in the context of diseases such as type II diabetes, cancer, obesity, diarrhoea, thrombosis, dental caries, immunodeficiency, and cardiovascular diseases including hypertension and hypercholesterolaemia [61].

BP also have multiple uses in food manufacturing. In addition to their valuable nutritional and therapeutic properties, BP can also retard oxidative degradation of lipids, thus improving the quality and nutritional values of foods [62].

BP are of interest as cosmetic ingredients due to their abilities to modulate cell proliferation, cell migration, inflammation, angiogenesis, melanogenesis, and protein synthesis. Both in vitro and in vivo studies demonstrate the possible applications of BP in dermatology, including wound healing, acne, skin pigmentation irregularities, and dermatitis [63].

### 5.1. Anti-Inflammatory Activity

Bioactive peptides from plant and animal sources have known anti-inflammatory activities (Figure 3). However, the mechanisms of action are mostly unknown and only a few of these peptides have been investigated with both in vitro and in vivo studies [64]. A study by Majumder et al. suggests that BP are anti-inflammatory because they cause transcriptional downregulation of inflammatory kinases such as NF-kB and MAPK. It is unknown whether BP act directly on cell membranes or by engaging cell receptors, but it is likely that both mechanisms play a role depending on the specific peptide [65]. Other suggested mechanisms of action include the inhibition of the pro-inflammatory JNK-MAPK pathway, reducing the formation of atherosclerotic plaques (obtained with IPP and VPP peptides) [66] and the modulation of the expression of intestinal chemokines and cytokines (obtained with beans, milk, and soy peptides) [67]. Several BP act through different pathways. Lunasin inhibits IL-6, IL-1β PGE2 production, the expression of COX-2 and inducible NOS, as well as the activation of NF-kB via the Akt-mediated route [68]. The polypeptide DMPIQAFLLYQEPVLGPVR derived from β-casein and a tripeptide derived from ovotransferrin from egg albumin, inhibit the NF-kB pathway, reducing the transcription of vascular cell adhesion molecule 1 (VCAM1) and intracellular adhesion molecule 1 (ICAM-1) [24,69]. The γ-glutamyl cysteine peptide extracted from beans inhibits JNK and IkB phosphorylation, while valyl-prolyl-tyrosine (VPY) reduces IL-8 and TNF-α secretion [64]. An interesting anti-inflammatory peptide has been extracted from milk fermented with *Lactobacillus plantarum* strains resulting in a BP with effects comparable to sodium diclofenac in preclinical studies [68]. BP derived from milk reduce postprandial inflammation in obese people as shown by reduced plasma monocyte chemoattractant protein-1 (MCP-1) and chemokine ligand 5 (CCL5) concentrations [70].

Although preliminary data suggest an interesting anti-inflammatory activity of BP related to the modulation of transcription factors and the inhibition of the expression of pro-inflammatory cytokines and chemokines, data on humans are still lacking and long-term RCTs are urgent to investigate and consolidate these results.

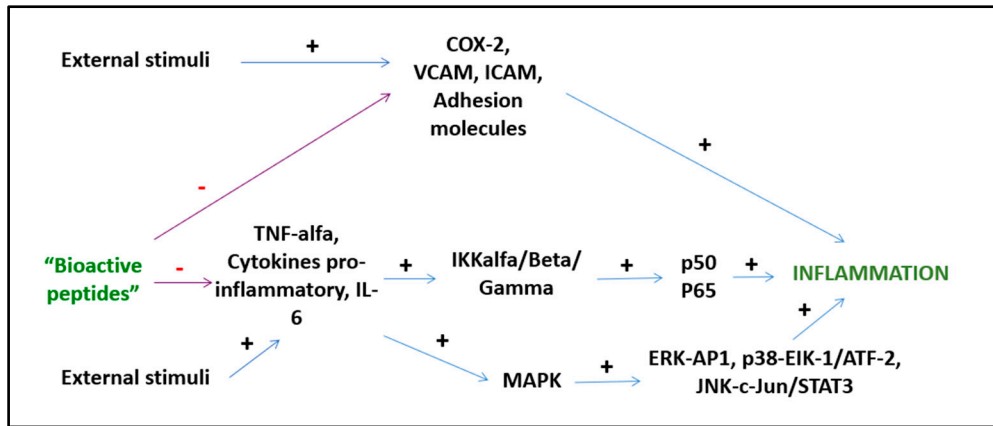

**Figure 3.** Anti-inflammatory activities of BPs.

### 5.2. Anti-Hypertensive Activity

Hypertension is one of the most important cardiovascular risk factors and data indicate that the lifetime risk of developing hypertension is a staggering 90%. The estimated global burden of hypertension will increase to 1.56 billion afflicted individuals by 2025 [70]. Hypertension accounts for about 7.6 million premature deaths and 92 million DAILYs (disability-adjusted life-years: 1 DAILY = 1 lost year of healthy life) [71]. In this context, the European guidelines for hypertension include the nutraceutical approach for both pre-hypertensive subjects with borderline blood pressure values and hypertensive patients in combination with conventional treatments [72,73].

Numerous bioactive peptides from different sources (milk, fish, plants, and meat) have demonstrated anti-hypertensive activities. They act through several pathways, such as inhibiting the renin-angiotensin system, increasing the activity of certain vasodilating agents (nitric oxide), or reducing the activity of the sympathetic nervous system [74]. The most common target of BP lowering molecules is the renin-angiotensin system, with the specific inhibition of renin or angiotensin converting enzyme (ACE) resulting in reduced levels of angiotensin I and angiotensin II [75]. One of the richest sources of anti-hypertensive peptides is milk. Milk is particularly rich in tripeptides (valyl-prolyl-proline (VPP) and isoleucyl-prolyl-proline (IPP)) and polypeptides (e.g., FFVAPFPEVFGK and YLGYLE-QLLR) [24]. Numerous RCTs have investigated the effects of milk BPs on blood pressure. In a meta-analysis of 18 RCTs, the lactotripeptides (LTP) IPP and VPP (dosages from 5 to 100 mg/day) reduced systolic blood pressure −3.73 mmHg (95% CI: −6.70, −1.76) and diastolic blood pressure −1.97 mmHg (95%CI: −3.85, −0.64) [76]. Interestingly, these effects were more evident in Asian people, suggesting a possible genetic/population-dependent effect. LTP also improves arterial stiffness, measured as pulse wave velocity, in mildly hypertensive subjects [77].

The enzymatic or pepsin hydrolysis of whey proteins generates several BP with anti-hypertensive activities. The decapeptide DRVYIHPFHL, octapeptide DRVYIHPF and heptapeptide RVYIHPF inhibit the renin-angiotensin-aldosterone (RAS) system [19]. Casein proteins and BP from cow's milk whey demonstrate significant blood pressure lowering activities in pre-hypertensive and hypertensive subjects [78]. The enzymatic hydrolysis of whey proteins also produces the ACE-inhibitor peptides α-lactalbumin and β-lactoglobulin, and lactorphins, which lower the blood pressure by normalizing the endothelial function [79].

Other anti-hypertensive tripeptides (LKP, IKP, LRP) are extracted from fish (e.g., bonito, tuna, sardine). These BP increase the endothelial NO levels and aorta vasodilatation in rats [80]. The peptides MVGSAPGVL and LGPLGHQ (from *Okamejei kenojei*), and AHIII (from *Styela clava*) exhibit anti-hypertensive activity in pre-clinical studies.

BP extracted from numerous plant species (e.g., soy, barley, oak, pea) appear to reduce blood pressure through several mechanisms; however, it is not always possible to

discriminate between the effects of plant proteins and other plant components that could contribute to the antihypertensive effects [81].

Therefore, BP derived from both plant and animal sources can moderately reduce blood pressure in humans. However, long-term RCTs, especially in normotensive and pre-hypertensive patients, are needed.

### 5.3. Lipid-Lowering Activity

Elevated plasma concentrations of total cholesterol (TC) and LDL cholesterol (LDL-C) and, under certain conditions, low concentrations of HDL cholesterol (HDL-C) are among the main modifiable risk factors for cardiovascular diseases [82]. An examination of data from over 18,000 individuals aged >20 years who participated in national health and nutrition surveys in the United States from 1999 to 2006 demonstrated that the unadjusted prevalence of hypercholesterolemia varies from 53.2% to 56.1%. Indeed, a recent report from the American Heart Association (AHA) has confirmed that in the United States, only 75.7% of children and 46.6% of adults have targeted TC levels (TC < 170 mg/dL for children and <200 mg/dL for adults, in subjects not treated pharmacologically) [83]. These percentages are comparable with other western countries [84,85].

BP from soy, lupine, and milk proteins are in the lipid-lowering nutraceuticals class. A meta-analysis of 35 RCTs investigated the effects of soy protein (B-conglycinin globulin) on cholesterolaemia. The results demonstrated the reduction in LDL-C of 3% ($-4.83$ mg/L; 95% CI: $-7.34$, $-2.31$), in TC of 2% ($-5.33$ mg/L; 95% CI: $-8.35$, $-2.30$) and in triacylglycerol of 4% ($-4.92$ mg/L; 95% CI: $-7.79$, $-2.04$). The period of treatment was from 4 weeks to 1 year and the lipid-lowering effects was greater in moderately hypercholesterolaemic patients ($-7.47$ mg/L; 95% CI: $-11.79$, $-3.16$) compared with healthy people ($-2.96$ mg/L; 95%CI: $-5.28$, $-0.65$) [86]. BP likely improve the lipid profile by acting as hydroxymethylglutaril-CoA (HMG-CoA) reductase inhibitors, up-regulators of LDL receptors, regulators of the sterol regulatory element-binding protein 2 (SREBP2) pathway, or by stimulating the faecal excretion of bile salts [87]. The hydrolysate extract of *Mucuna pruriens* and BP from cowpea reduced LDL-C and TC by affecting micelle formation and exogenous cholesterol absorption [88,89]. The lunasin peptide extracted from soy has demonstrated lipid-lowering activities in animal models [90].

Therefore, several lipid-lowering peptides can improve the lipid profiles of mildly dyslipidaemic patients. However, evaluation of both pharmacodynamic and pharmacokinetic profiles and the long-term effective dosages in humans are necessary.

### 5.4. Anti-Cancer Activity

One of the most important areas of research for bioactive peptides concerns their role as anti-cancer agents. BP from plants, milk, egg, and marine organisms possess cytotoxic activity against numerous cancer cell lines. These BP have the advantages of low-toxicity, and high tissue penetration, cell diffusion, and permeability [91]. BP can use different mechanisms to inhibit cell migration, affect gene transcription/cell proliferation, inhibit tumor angiogenesis, or alter cancer cell tubulin structures [92].

The BP lunasin has anticancer activity when tested against breast, skin, colon, prostate, leukaemia, and lymphoma cell lines [93]. Lunasin suppresses the transformation of cells by chemical carcinogens (in mouse fibroblast, and human breast cancer MCF-7), induces cell cycle arrest in G2/M phase and apoptosis through the activation of caspase-3 (in L1210 leukaemic and human colon adenocarcinoma cells) and inhibits the metastasis of human colon cancer cells [94]. In vivo studies, especially in mice models, demonstrated the reduction of lymphoma volume and liver metastasis of colon cancer by lunasin [95].

The peptide Glutammyl-Glycyl-Argininyl-Prolyl-Arginine from rice, at the dose of 600–700 µg/mL, inhibits the growth of colon cancer cells (Caco-2 and human colorectal adenocarcinoma cell line, HCT-116) by 84%, breast cancer cells (MCF-7, MDA-MB-231) by 80%, and liver cancer cells by 84% (HepG-2) [96].

Other rich sources of anticancer BP are the legume seeds that contain protease inhibitors, such as the Bowman-Birk inhibitor that has an inhibitory effect against prostate, breast, and colon cancers in vitro [97]. The Bowman-Birk inhibitor has Food and Drug Administration (FDA) approval, especially for people with oral leukoplakia or benign prostatic hyperplasia [98]. Plant-derived lectins (from tepary bean and mistletoe) have cytotoxic effects on the cervical carcinoma cell line C33-A and human colon carcinoma cell line, the Sw480 [99].

Among the BP of animal origin, milk lactoferrin inhibits the growth of breast cancer (MDA-MB-231) and nasopharyngeal carcinoma cells in vitro and in vivo by arresting the cell cycle at the G1/S transition, or suppressing Akt signaling, respectively [100]. It also induces cell apoptosis, modulates gene expression and reduces angiogenesis [101]. Hydrolysates of the egg proteins lysozyme and ovomucin inhibit tumor cell proliferation, improving the effectiveness of chemotherapy (colorectal cancer; B16 melanoma) [102]. Many marine peptides (e.g., from tuna, sponges, squid) show antiproliferative activity against different cancer types in vitro, but these results have not been confirmed in vivo [103].

Thus, several BP demonstrate potential as anti-cancer peptides, with demonstrated cytotoxic and anti-tumoural activity in vitro and in animal models. However, controlled human clinical trials are still needed to evaluate the true therapeutic efficacy of this class of compounds.

### 5.5. Immunomodulatory Activity

BP can also modulate immune responses. Studies in vitro and in vivo demonstrated that αS1-casein and β-casein stimulated phagocytes, B lymphocyte IgG production, and T cell proliferation [24]. Peptides from fish have immunomodulatory activities, especially enhancing macrophage and natural killer cells activity, and lymphocyte proliferation (e.g., BP from Chum Salmon or from Atlantic cod) [104]. BP can also regulate immunity by modulating the gut-associated immunity affecting phagocytic cell activity and IgA-secreting cells in the small intestine lamina propria [105]. Studies in mice demonstrated that oyster hydrolysates can enhance lymphocyte proliferation, natural killer cell activity, and macrophage phagocytosis [106]. Similar results were observed with tryptic hydrolysates of soybean and rice proteins [107].

Thus, BP exhibit immunomodulatory effects on both innate and adaptive immunity. RCTs are clearly needed to determine the efficacy and the safety profile of these molecules.

### 5.6. Other Biological Activities

BP from wheat gliadin, soy proteins, egg-yolk proteins, porcine myofibrillar proteins, pea, and aquatic by-product proteins are well known to possess antioxidant properties and to protect against the oxidative stress typical of major chronic diseases [108,109].

BP antioxidant activity can be a result of the metal ion chelation that inhibits enzymatic and non-enzymatic peroxidation of lipids and essential fatty acids. BP can also reduce reactive oxygen species by acting as free radical scavengers. Carnosine and anserine, two abundant BP in meats, reduce or prevent oxidative stress-related diseases [28]. Lunasin protects human Caco-2 cells in vitro from oxidative stress caused by treatment with hydrogen peroxide or tert-butylhydroperoxide by scavenging peroxyl and superoxide radicals [110]. BP isolated from oyster, shrimp, squid, and bluemussel have demonstrated antioxidant properties in vitro and in animal models [111].

The peptides α-lactorphin and β-lactorphin (opioid peptides) exert analgesic activity on the opiate receptor agonist. In in vitro studies, α-lactalbumin and β-lactoglobulin BP have demonstrated analgesic activity at micromolar concentrations [112]. Analgesic peptides have also been isolated from rice albumin, gluten, and spinach [113].

The BP lactoferrin (fragment 17–41) and peptides derived from bovine meat (GFHI, DFHING, FHG, GLSDGEWQ) are being studied in vitro and in vivo for their antimicrobial effects. BP have demonstrated a large spectrum of action against viruses, bacteria, protozoa, and fungi. For example, the fragment 17–41 of lactoferrin, also known as lactoferricin,

has bactericidal activity due to its ability to alter bacterial membrane permeability by interacting with the lipid A portion of bacterial lipopolysaccharides [114]. Preliminary data indicate that the BP are well tolerated, do not induce pathogen resistance, and demonstrate activity against both Gram-positive and Gram-negative bacteria [115].

## 6. Discussion

The interest in BP is growing year by year, owing to current progress in clinical trials and efforts of the food and nutraceutical industries [116]. Many BP are well known to regulate different aspects of cellular function and communications [60].

Several studies conducted in human populations have demonstrated the effectiveness of BP, however, some limitations should be considered. These clinical trials often have small sample sizes, short treatment periods, and highly variable results [117]. Currently, the best evidence in humans is for supplementation with BP in cardiovascular prevention, in particular as lipid-lowering or anti-hypertensive agents [24]. BP appear to be valid nutraceutical options if combined with other molecules, confirming their efficacy, tolerability and safety profile [75]. The antioxidant, anti-inflammatory, and analgesic activities of BP have been investigated in a variety of studies, but long-term clinical trials are still lacking and their mechanisms of action need clarification [19]. Interesting preliminary data demonstrate the potential of BP as anti-cancer applications, but more phase I studies are needed [116]. Finally, the immunomodulatory actions of different peptides on both innate and adaptive immunity are under investigation in pre-clinical studies [24].

The available data on BP present limitations beyond the paucity of clinical trials. We often lack the pharmacokinetic data required to determine the dosage, frequency of administration, and to understand the different inter- and intra-individual variability [118]. The inter-individual variability (age, sex, diseases, concomitant therapy etc.) of BP effects needs to be considered. For example, lactotripeptides (IPP and VPP) showed differential efficacy for blood pressure reduction in European and Asian subjects, suggesting a genetic/population-dependent effect [119].

Pharmacokinetic studies are important to study the biopharmaceutical aspects of BP and consequently to develop the best formulative strategies (e.g., lipid microparticle systems, micelle, emulsion, microencapsulation etc.) to improve oral bioavailability and bioactivity [76]. However, these studies are often limited because BP have a short half-life (less than 2 h) as well as a low plasma concentration (pmol/mL), thus limiting the measurement of oral bioavailability [120].

Oral bioavailability of BP is typically extremely low and the chemical structure of peptides must be considered. In this context, in vitro studies using simulated gastrointestinal digestion systems are needed determine the stability and bioaccessibility of many BP [121]. In fact, digestive processes can inactivate or reduce the activities of BP, or convert the peptides into forms unable to reach the bloodstream in sufficient quantities to exert their effects. Recent development of non-conventional dosage forms has improved BP bioavailability, reducing both microbiota and chemical degradation, the main limitations for oral peptide supplementation [47].

More data on the pharmacodynamics of BP are also needed. The putative mechanisms of actions are often unclear and individual BP appear to act through several pathways and possess several pleiotropic activities. It is also difficult to attribute the action to a single peptide when the entire complex of protein hydrolysates is studied [122]. A possible solution might be in vitro activity guided fractionation, characterized by the combination of the analytical separation of digested protein fractions, followed by the in vivo evaluation of specific BP activity [123].

Another important aspect requiring further investigation is the determination of the different transporters involved in the intestinal absorption of BP. The specific duodenal transporter that absorbs the majority of BP can be saturated, depending on the dosages. Thus, competition between two or more peptides for the same transporter can reduce the bioavailability [123].

## 7. Conclusions

BP studies are encouraging and demonstrate their therapeutic potential in nutraceuticals and food supplements. BP can prevent and/or treat (in addition to conventional therapies) several diseases and/or risk factors. However, pharmacokinetic and bioactivity human studies are urgently needed to better understand these compounds. Long-term, randomized clinical trials are also needed to test their efficacy and potential immunogenicity. Finally, the isolation, extraction and production techniques of BP must be standardised and made scalable to industrial applications. In this context, an economic analysis of the potential therapeutic use of BP must be considered.

**Author Contributions:** The authors equally co-worked to the paper. All authors have read and agreed to the published version of the manuscript.

**Funding:** This research received no external funding.

**Institutional Review Board Statement:** Not applicable.

**Informed Consent Statement:** Not applicable.

**Data Availability Statement:** Not applicable.

**Conflicts of Interest:** The authors declare no conflict of interest.

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
