# Peer review of "Pharmacodynamics and Clinical Implications of the Main Bioactive Peptides: A Review"

_nutraceuticals, doi:10.3390/nutraceuticals2040030_

Round 1

Reviewer 1 Report

1. The author needs to add few clinical studies on bioavaliability of BPs from food sources.

2. Put sequences of BPs found form milk or plants in Table format with bioavailabilities.

3. Mechanisms of Bioavailabilty of peptides needs to be explained with Graphical representation to make the mechanism more clear.

Author Response

The authors consider very important the bioavailability data inherent to the BPs. However, to our knowledge, the pharmacokinetic studies available to date on most BPs are either lacking or severely limiting in study strength. Therefore, the authors cited information related to the bioavailability of BPs only when the bibliographic information allowed it. Certainly, this is an aspect in which a lot of work will have to be done to clarify the biopharmaceutical fate of these assets. Anyway, we have modified and improved the text, when possible.

Reviewer 2 Report

The authors summarize information about bioactive peptides from the last two decades regarding sources, stability, bioavailbility and applications.

The review is well researched like other, similiar reviews about bioactive peptides. But in my opinion the uniqueness is lacking and I'm looking for the connection to nutraceuticals products. In line 224: Nutraceuticals are molecules of either animal or plant origin that exert a positive impact on human health when incorporated into food or pharmaceutical formulations (sachets, capsules, tablets, etc.). What kind of products use BP in more detail?

What about bioactive peptides as nutraceuticals regarding supplementation?

Line 174: "New separation and purification techniques (well described in the next chapters) have been developed to overcome these problems." I cannot find them? Only some purification description in general.

Line 423: "Many BP peptides" only BP?

Author Response

The authors summarize information about bioactive peptides from the last two decades regarding sources, stability, bioavailbility and applications. The review is well researched like other, similiar reviews about bioactive peptides. But in my opinion the uniqueness is lacking and I'm looking for the connection to nutraceuticals products. In line 224: Nutraceuticals are molecules of either animal or plant origin that exert a positive impact on human health when incorporated into food or pharmaceutical formulations (sachets, capsules, tablets, etc.). What kind of products use BP in more detail?

Authors: As explained in the introduction, to date, the European regulation provides that there may be an overlap between some molecules that can be commercialized as both drugs and food supplements. This is also the case with most bioactive peptides (for example IPP and VPP for the treatment of mild hypertension, or BP from bromelain for the reduction of oedema in different inflammatory diseases). In this regard, an increasingly close dialogue with national and European regulatory authorities is certainly necessary in order to strictly regulate and define the field of nutraceuticals, including also BP.

Line 174: "New separation and purification techniques (well described in the next chapters) have been developed to overcome these problems." I cannot find them? Only some purification description in general.

Authors: Thank you for the suggested correction. We changed the text accordingly.

Line 423: "Many BP peptides" only BP?

Authors: Thank you for the suggested correction. We changed the text accordingly.

Reviewer 3 Report

The manuscript entitled " Pharmacodynamics of the main bioactive peptides: clinical implications " presents a systematic review mainly concerning the introduction to bioactive peptides, describe their production and application methods, present data on bioactivity and bioavailability and finally highlight the perspectives of this class of molecules in the clinical setting. Although well written and the arguments presented clearly, there are several problems that do not allow immediate publication. Authors are requested to extensively review the work underlying the following indications:

1)      The title should be changed as it does not fully reflect the content of the work. Clearly indicate that this is a "review”.

2)      Keywords are missing.

3)      It is not correct to refer to a "chapter" if you are writing a review (line 57)

4)      Table 1 is not clear. Tables and figures should be understandable even beyond the text. Provide more information also in the caption.

5)      Is there national, European, etc ... legislation that specifically concerns BPs? Possibly insert in the introduction.

6)      Line 190: specify the abbreviation "GI"

7)      In describing the anti-inflammatory properties (line 243), it would be much more useful to accompany the description with diagrams or figures that make the understanding more immediate.

8)      Lines 479-480: it is not necessary to add this sentence.

9)      The conclusions refer to the need to discuss the production of BPs from an economic point of view. Yet such a discussion or a draft of it was never made during the manuscript. It would be useful to add it.

10)  Where have the new strategies of separation and purification mentioned in lines 174-175 been explored?

11)  Are there no studies reporting any adverse human health effects resulting from the use of BPs?

12)  In reviews it is often useful to accompany the text with summary tables in which it is possible to summarize many other studies without the need to describe them in the text. It is advisable to implement the different parts of the review with tables. For example, a table can be made indicating the sources, effects, production methods and relative references.

13)  Some references are really dated. Given the topicality of the argument that the authors emphasize, it would be necessary to replace older references with more recent studies. To this end, please also include this very interesting work:

Nongonierma, Alice B., et al. "Release of dipeptidyl peptidase IV (DPP-IV) inhibitory peptides from milk protein isolate (MPI) during enzymatic hydrolysis." Food Research International 94 (2017): 79-89.

14)  Figure 2: This figure needs to be revised and improved. Unfortunately, understanding is severely limited by size and quality. Please arrange.

Author Response

The manuscript entitled " Pharmacodynamics of the main bioactive peptides: clinical implications " presents a systematic review mainly concerning the introduction to bioactive peptides, describe their production and application methods, present data on bioactivity and bioavailability and finally highlight the perspectives of this class of molecules in the clinical setting. Although well written and the arguments presented clearly, there are several problems that do not allow immediate publication. Authors are requested to extensively review the work underlying the following indications:

  • The title should be changed as it does not fully reflect the content of the work. Clearly indicate that this is a "review”.

Authors: The title has now been added as suggested by the reviewer.

2)      Keywords are missing.

Authors: Keywords have now been added as suggested by the reviewer.

  • It is not correct to refer to a "chapter" if you are writing a review (line 57)

Authors: Thank you for your suggested correction.

  • Table 1 is not clear. Tables and figures should be understandable even beyond the text. Provide more information also in the caption.

Authors: The authors provide now more information in the text regarding the role of table 1.

  • Is there national, European, etc ... legislation that specifically concerns BPs? Possibly insert in the introduction.

Authors: As explained in the introduction, to date, the European regulation provides that there may be an overlap between some molecules that can be commercialized as both drugs and food supplements. This is also the case with most bioactive peptides. In this regard, an increasingly close dialogue with national and European regulatory authorities is certainly necessary in order to strictly regulate and define the field BPs and its regulatory area.

6)      Line 190: specify the abbreviation "GI"

Authors: Thank you for your suggested correction.

7)      In describing the anti-inflammatory properties (line 243), it would be much more useful to accompany the description with diagrams or figures that make the understanding more immediate.

Authors: A specific figure has now been inserted in order to make more immediate the action of BPs on inflammation.

8)      Lines 479-480: it is not necessary to add this sentence.

Authors: The sentence has been removed, as suggested.

9)      The conclusions refer to the need to discuss the production of BPs from an economic point of view. Yet such a discussion or a draft of it was never made during the manuscript. It would be useful to add it.

Authors: The authors agree with the reviewer: however, the authors wanted to highlight an aspect that deserves separate treatment (this is not the main object of the review) by people specifically competent in nutra-economics.

11)  Are there no studies reporting any adverse human health effects resulting from the use of BPs?

Authors: To date, the supplementation of BPs resulted safe and well tolerated despite that few long-term RCTs have been conducted until now.

12)  In reviews it is often useful to accompany the text with summary tables in which it is possible to summarize many other studies without the need to describe them in the text. It is advisable to implement the different parts of the review with tables. For example, a table can be made indicating the sources, effects, production methods and relative references.

Authors: We usually agree with the reviewer. In this context, the problem is that literature is largely heterogeneous, so that a large number of small (but huge) tables should occupy the text, complicating the text reading.

13)  Some references are really dated. Given the topicality of the argument that the authors emphasize, it would be necessary to replace older references with more recent studies. To this end, please also include this very interesting work:

Nongonierma, Alice B., et al. "Release of dipeptidyl peptidase IV (DPP-IV) inhibitory peptides from milk protein isolate (MPI) during enzymatic hydrolysis." Food Research International 94 (2017): 79-89.

Authors: We are in agreement with the reviewer and the reference suggested as now been inserted in the text.

14)  Figure 2: This figure needs to be revised and improved. Unfortunately, understanding is severely limited by size and quality. Please arrange.

Authors: Figure 2 has been now revised and improved as suggested by the reviewers.

Reviewer 4 Report

The manuscript is devoted to reviewing the literature on biologically active peptides (BP) as nutraceuticals, the sources from which they are extracted, methods of isolation, specific biological activities of BP, discussing the possibility of their oral application and ways to increase bioavailability. The list of quoted literary sources consists of 145 titles, so the review fully reveals the stated topic. However, it would be desirable to draw a clearer line between peptide pharmaceuticals and peptide food additives. This would greatly facilitate the perception of the data presented.

Some brief comments:

1.      Lines 261, 288-289, 358: According to recommendations of IUPAC-IUB Joint Commission on Biochemical Nomenclature (Pure & Appi. Chem., Vol. 56, No. 5, pp. 595—624, 1984) to name peptides, the names of acyl groups ending in 'yl’ are used. Only the C-terminal residue is represented by the name of the amino acid. So, correct peptide names and formulas are valyl-prolyl-tyrosine, valyl-prolyl-proline, etc.

2.      Fig.2: Prodrug approach – phenyl propionic acid, coumarinic acid and acyloxyalkoxy moiety were use for synthesis of cyclic prodrugs of opioid peptides (J Control Release. 1999 Nov 1;62(1-2):231-8). Another strategy for prodrug synthesis have focused on modification of a single functional group (DDT.1997v.2(4):148-55). Advantages and drawbacks should be corrected.

3.      Fig.2: PEGylation – PEG (macrogol) is a polyethylene glycol, not a polyethylglycol.

4.      Fig.2: N-methylation - In order to increase enzymatic stability of peptides, methyl group can be attached to amino group of any amino acid residue included in a cleavage site. Advantages and drawbacks of the approach should be corrected.

Author Response

The manuscript is devoted to reviewing the literature on biologically active peptides (BP) as nutraceuticals, the sources from which they are extracted, methods of isolation, specific biological activities of BP, discussing the possibility of their oral application and ways to increase bioavailability. The list of quoted literary sources consists of 145 titles, so the review fully reveals the stated topic.

Authors: We are grateful for the reviewer’ positive comment on our work.

However, it would be desirable to draw a clearer line between peptide pharmaceuticals and peptide food additives. This would greatly facilitate the perception of the data presented.

Authors: The authors agree with the reviewer; however, to date, the European regulation provides that there may be an overlap between some molecules that can be commercialized as both drugs and food supplements. This is also the case with most bioactive peptides. In this regard, an increasingly close dialogue with national and European regulatory authorities is certainly necessary.

Some brief comments:

  1. Lines 261, 288-289, 358: According to recommendations of IUPAC-IUB Joint Commission on Biochemical Nomenclature (Pure & Appi. Chem., Vol. 56, No. 5, pp. 595—624, 1984) to name peptides, the names of acyl groups ending in 'yl’ are used. Only the C-terminal residue is represented by the name of the amino acid. So, correct peptide names and formulas are valyl-prolyl-tyrosine, valyl-prolyl-proline, etc.

Authors: Thank you for the suggested correction. We changed the text accordingly.

  1. 2: Prodrug approach – phenyl propionic acid, coumarinic acid and acyloxyalkoxy moiety were use for synthesis of cyclic prodrugs of opioid peptides (J Control Release. 1999 Nov 1;62(1-2):231-8). Another strategy for prodrug synthesis have focused on modification of a single functional group (DDT.1997v.2(4):148-55). Advantages and drawbacks should be corrected.

Authors: Thank you for the suggested correction. We changed the text accordingly.

  1. 2: PEGylation – PEG (macrogol) is a polyethylene glycol, not a polyethylglycol.

Authors: Thank you for the suggested correction. We changed the text accordingly.

Fig.2: N-methylation - In order to increase enzymatic stability of peptides, methyl group can be attached to amino group of any amino acid residue included in a cleavage site. Advantages and drawbacks of the approach should be corrected.

Authors: Thank you for the suggested correction. We changed the text accordingly.

Round 2

Reviewer 3 Report

While nearly all the proposed revisions have been performed by the authors, two issues remain:

- Pay attention to the final part of the conclusions section. The lines to be deleted are 492-493 and not lines 490-491.

- The bibliography is not updated. There are still dated references (e.g. 2, 32, 42, 51, 58, 59, 60, 115, 117). Why is it not possible to replace them with more recent ones that perhaps contain the same type of information?

Author Response

We are grateful to the reviewer for the suggestion!

- Pay attention to the final part of the conclusions section. The lines to be deleted are 492-493 and not lines 490-491.

A: We deleted the lines 492-493, as per reviewer suggestions.

- The bibliography is not updated. There are still dated references (e.g. 2, 32, 42, 51, 58, 59, 60, 115, 117). Why is it not possible to replace them with more recent ones that perhaps contain the same type of information?

A: We did our best to update cited literature, as per reviewer suggestion.